# Higher hypnotic suggestibility is associated with the lower EEG signal variability in theta, alpha, and beta frequency bands

**Soheil Keshmiri**[1]*, **Maryam Alimardani**[1,2], **Masahiro Shiomi**[1], **Hidenobu Sumioka**[1], **Hiroshi Ishiguro**[1,3], **Kazuo Hiraki**[4]

**1** Advanced Telecommunications Research Institute International (ATR), Kyoto, Japan, **2** Department of Cognitive Science and Artificial Intelligence, Tilburg University, the Netherlands, **3** Graduate School of Engineering Science, Osaka University, Japan, **4** Department of General Systems Studies, Tokyo University, Japan

* soheil@atr.jp

**Data Availability Statement:** The files containing the DE features extracted from EEG signals in each electrode are made available in the supporting information.

## Abstract

Variation of information in the firing rate of neural population, as reflected in different frequency bands of electroencephalographic (EEG) time series, provides direct evidence for change in neural responses of the brain to hypnotic suggestibility. However, realization of an effective biomarker for spiking behaviour of neural population proves to be an elusive subject matter with its impact evident in highly contrasting results in the literature. In this article, we took an information-theoretic stance on analysis of the EEG time series of the brain activity during hypnotic suggestions, thereby capturing the variability in pattern of brain neural activity in terms of its information content. For this purpose, we utilized differential entropy (DE, i.e., the average information content in a continuous time series) of theta, alpha, and beta frequency bands of fourteen-channel EEG time series recordings that pertain to the brain neural responses of twelve carefully selected high and low hypnotically suggestible individuals. Our results show that the higher hypnotic suggestibility is associated with a significantly lower variability in information content of theta, alpha, and beta frequencies. Moreover, they indicate that such a lower variability is accompanied by a significantly higher functional connectivity (FC, a measure of spatiotemporal synchronization) in the parietal and the parieto-occipital regions in the case of theta and alpha frequency bands and a non-significantly lower FC in the central region's beta frequency band. Our results contribute to the field in two ways. First, they identify the applicability of DE as a unifying measure to reproduce the similar observations that are separately reported through adaptation of different hypnotic biomarkers in the literature. Second, they extend these previous findings that were based on neutral hypnosis (i.e., a hypnotic procedure that involves no specific suggestions other than those for becoming hypnotized) to the case of hypnotic suggestions, thereby identifying their presence as a potential signature of hypnotic experience.

**Funding:** This research was supported by Japan Society for the Promotion of Science (JSPS), Grant-in-Aid for JSPS Research Fellow, and Japan Science and Technology Agency (JST). S.K. was funded by JSPS, KAKENHI (JP19K20746) and JST, CREST (JPMJCR18A1). M.A. was funded by Grant-in-Aid for JSPS Research Fellow (15F15046) and JST, CREST (2014-PM11-07-01). M.S. was funded by JST, CREST (JPMJCR18A1). K.H. was funded by JST, CREST (2014-PM11-07-01). The funders had no role in study design, data collection and analysis, decision to publish, or preparation of the manuscript.

**Competing interests:** The authors declare that the research was conducted in the absence of any commercial, financial, and/or non-financial relationships that could be construed as a potential conflict of interest.

## Introduction

Hypnosis has received a growing interest from cognitive neuroscience research due to its utility for not only advancing our understanding of the state of consciousness [1, 2] but also as a potential tool in treatment of a number of chronic and psychological disorders [3–7]. Oakley et al. [8] define hypnosis as a change in baseline mental activity in response to induction and/or a set of verbal instructions (referred to as suggestions) that facilitate such hypnotic mental states as increased absorption, focused attention, and reduced spontaneous thoughts [9]. Although a typical hypnotic phenomenon (e.g., sensory experience, amnesia, etc.) requires specific suggestions, research indicates that hypnotizability is rather associated with the brain activity during attention outside hypnosis [10]. In other words, individuals are able to respond to hypnotic suggestions without the need for a formal induction procedure [8, 11]. In fact, Braffman and Kirsch [12] consider this responsiveness outside the hypnotic state as a predictor of suggestibility of individuals during hypnosis. A review of the literature by Gruzelier [13] provides further support for the correspondence between attentional capability of individuals and their degree of suggestibility during hypnosis.

The study of hypnotic state using EEG [14, 15] presents a promising gateway for assessing the effect of hypnosis on the brain neural activity. This is due to the mind-brain supervenience [16] conjecture which states that the mental and cognitive events are accompanied by changes at neural level. It is apparent that the ability to infer such a correspondence between the mental and cognitive events on the one hand and the change in the neural activity that accompanied them on the other hand can provide a robust basis for realization of the neurophysiological [1, 2, 15, 17, 18] and socio-psychological bases of hypnosis phenomenon [19–22]. Such an understanding can also help realize the potential of hypnosis as a solution concept for clinical treatment of mental and behavioural disorders at brain functional level [23–26].

Although a number of previous studies reported a significant change in spectral band power between pre- and post-hypnotic induction and/or high and low suggestible individuals [27, 28–32], these findings appeared to be inconclusive [33, 34]. For instance, whereas some pointed at an increase in the theta activity in high hypnotizable subjects [35, 36], others reported on its reduced [37] or even absence [27] of activity. Such inconsistencies can be attributed to the use of a direct measure of band amplitude (e.g., averaging over a given spectral power) to quantify the effect of hypnotic experience on the human subjects' brain activity. Wutz et al. [38] pointed that the modulation of information does not necessarily involve change in local power, thereby implying the possibility of the presence of a significant information when power is not elevated. Moreover, Jamieson and Burgess [39] stated that given the equivalent sensory and behavioural processing demands in pre- and post-hypnotic phases, it is not reasonable to expect a significant difference in the spectral band amplitude between these settings and/or the brain activity of high and low hypnotizable individuals. These results, collectively, identified that the mere changes in the band amplitude did not represent a plausible measure for analysis of the potential effect of hypnotic experience on individuals' brain activity [40].

To address this shortcoming, a number of EEG-based biomarkers for study and analysis of hypnosis phenomenon in human subjects have been introduced [14, 28, 39, 41]. For instance, Fingerkurts and colleagues [14] considered the structural synchrony measure [14] in the study of the neutral hypnosis (i.e., a hypnotic procedure that involves no specific suggestions other than those for becoming hypnotized) of a single hypnotic virtuoso. Their results indicated that this measure was able to detect the change in local and remote functional connectivity (FC) between the brain regions during hypnotic state. Terhune et al. [41] reported a significant reduction of Phase Lag Index among highly hypnotizable individuals which was more pronounced between the frontal and the parietal electrode groupings in the upper alpha band.

Cardeña et al. [29] found a relation between depth of hypnosis and the topographic variability in the beta and gamma bands. Jamieson and Burgess [39] utilized the coherence (COH) [42] and the imaginary component of coherence (iCOH) [43] to show an increase in the theta and a decrease in the beta1 (13.0-19.9 Hz) band from pre-hypnosis to hypnosis condition among highly hypnotizable participants.

Although these measures provided encouraging results in identifying an EEG-based hypnotic biomarker, their applicability appeared to be limited. For instance, Deivanayagi et al. [44] found that COH associated the state of hypnosis with lowered theta and alpha frequency bands. They further envisioned the use of this measure to study the effect of hypnosis on higher frequencies such as beta and gamma bands. In contrast, Sabourin et al. [30] found that COH indicated an increase in theta power during hypnosis in both low as well as high hypnotizable individuals. They further observed that the change in alpha power was not a predictor of hypnotic susceptibility, that highly susceptible subjects had more beta activity in the left than right hemispheres, and that low susceptible subjects showed only a weak lateralized asymmetry. On the other hand, the structural synchrony measure [14] was only tested on a single hypnotic virtuoso and in a neutral hypnosis setting. This made it difficult to draw an informed conclusion on its utility in a broader domain (e.g., its sensitivity and specificity in a larger mixed group of high and low hypnotizable subjects and/or neural activity during hypnotic suggestions). In the same vein, the approach by Jamieson and Burgess [39] required to employ two different measures (i.e., COH and iCOH) for analysis of two frequency bands (i.e., theta and beta1, respectively). It is also worthy of note that their results did not identify any significant differences in power [39]. Furthermore, these results were primarily based on the state of hypnosis (i.e., without observing responses of the participants to hypnotic suggestions). It is apparent that a robust EEG-based hypnosis biomarker that exhibits a high specificity allows for drawing a more informed conclusion on the effect of hypnosis on the brain activity. Such a measure can provide adequate answers to divided perspectives on phenomenological [1, 2] as well as the role of hypnosis in clinical treatment of mental disorders [15, 17–26].

An important characteristic that is attributed to the brain functioning is the relation between the variation in the brain activity and its information content [45, 46]. For instance, Miller [47, p. 81] argued that there is a direct correspondence between the "amount of information" and the variance since "anything that increases the variance also increases the amount of information" (ibid.). Similarly, Cohen et al. [48] considered the ability to identify meaningful variation in the brain activation to be an indicator of an effective analysis approach. Accordingly, Lundqvist et al. [49] showed that the change in variation in information of neural spike rate best represents the burst of brain activity in response to working memory (WM) tasks. These findings that are in line with the concept of entropy [50], as originally formulated by Shannon [51], indicated the adequacy of the use of information-theoretic measures as summary statistics of the brain activity. In fact, it comes as no surprise that entropy in its various formulation [52] is utilized exhaustively for analysis of the information content of biological signals [53–55].

In the context of EEG time series analysis, DE appears to be first utilized by Duan et al. [56]. Subsequently, Zheng and Lu [57] noted the DE's ability to discriminate between EEG pattern of low and high frequency, given the EEG's higher low frequency energy over high frequency energy. They further showed (ibid.) that DE can outperform such features as differential asymmetry (DASM), rational asymmetry (RASM), and power spectral density (PSD) in EEG frequency-domain analysis. Shi et al. [58] used DE in the analysis of the EEG time series associated with vigilance. Alimardani et al. [59] also utilized DE to achieve a significantly above average classification accuracy of low versus high suggestible participants during hypnosis. In this respect, Keshmiri et al. [60] demonstrated that DE quantifies the information

content of brain activity in terms of a shift (e.g., increase and/or decrease) in its neural population spiking (i.e., its variation in information) as it is charactrized by Fano factor [61].

Given these findings, we sought the utility of DE for quantification of the brain neural responses to hypnotic suggestions. Specifically, we utilized DE of the theta, alpha, and beta frequency bands of fourteen-channel EEG recordings of twelve carefully selected high and low hypnotically suggestible individuals. We found that the higher hypnotic suggestibility was associated with a significantly lower variability in information content of theta, alpha, and beta frequencies. We also observed that such a lower variability was accompanied by a significantly higher functional connectivity (FC, a measure of spatiotemporal synchronization) in the parietal and the parieto-occipital regions in the case of theta and alpha frequency bands and a non-significantly lower FC in the central region's beta frequency band.

Our results contribute to the field in two ways. First, they identify the applicability of DE as a unifying measure to reproduce the similar observations that are separately reported through adaptation of different hypnotic biomarkers in the literature. Second, they extend these previous findings that were based on neutral hypnosis (i.e., a hypnotic procedure that involves no specific suggestions other than those for becoming hypnotized) to the case of hypnotic suggestions, thereby identifying their presence as a potential signature of hypnotic experience.

## Materials and methods

### Subjects

Forty-six subjects (17 females, age M = 24.2, SD = 6.4) participated in this experiment from which two were removed for not following the instructions properly. All participants were university students/staff and were right-handed. Eleven participants had previously experienced hypnosis either in form of a stage show or a research experiment. Participants received explanation prior to the experiment and signed a written informed consent form (Approval number: 412-2, University of Tokyo).

### Hypnosis test and suggestibility score

The experiment included a pre-recorded Harvard Group Scale of Hypnotic Susceptibility, Form A (HGSHS:A, referred to as Harvard test hereafter) [62]. It was administered for two purposes: 1) to test the subjects' susceptibility to hypnosis, and 2) to give a full hypnosis session for EEG recording. This test comprises of twelve items. They are: 1) Head falling 2) Eye Closure 3) Hand lowering 4) Arm immobilization 5) Finger lock 6) Arm rigidity 7) Hands moving 8) Communication inhibition 9) Fly hallucination 10) Eye catalepsy 11) Post-hypnotic amnesia 12) Post-hypnotic suggestion (touching left ankle). In addition to these, two more items "Cooling of hands" and "Warming of hands" were added before items 11 and 12. These items were prepared by a professional hypnotist and added to the instructions, immediately following item 10.

From twelve items in Harvard test, subjects with scores 0 through 3 were categorized in LOW suggestible group. Similarly, we included the subjects with scores 8 through 12 in High suggestible group. This resulted in 8 LOW participants and 6 High participants. The remainder of participants were considered as Mid suggestible group and were subsequently excluded from present analyses.

### Experimental procedure

After receiving explanation, subjects were seated in a comfortable chair and the experimenter placed the EEG electrodes.

Subjects were asked to avoid unnecessary movements during the recording unless they were instructed so. The recording took place in five stages (Fig 1a). It started by a five-minute baseline recording (Pre-baseline) with eyes open. Next, subjects listened to an audio file of hypnotic instructions (in Japanese). Instructions started with items one and two of Harvard test, corresponding to preparation and induction phases, respectively. The Induction phase (i.e., eye closure in Harvard test) lasted for fifteen minutes. It primarily included verbal instructions to help subjects enter a state of deep relaxation and focused attention. In the suggestion phase, subjects listened to items three through ten of Harvard test (i.e., hand lowering, arm immobilization, finger lock, arm rigidity, hands moving, communication inhibition, fly hallucination, and eye catalepsy) (Fig 1b), followed by two additional suggestions of cooling and warming. Each suggestion lasted for two to five minutes. The experimenter noted down the behavioural responses of subjects to each suggestion as the session progressed. Then, subjects entered the awake phase after which items eleven and twelve of Harvard test were administered, thereby bringing subjects back to alert condition. The entire hypnosis session lasted for fifty minutes. At the end of the session, subjects answered to Harvard scoring assessment questionnaire. Finally, we recorded a five minutes post-hypnosis baseline with eyes open.

## Data acquisition

EEG signals were recorded from 14 sites that covered the frontal, central, temporal, parietal and occipital areas. Electrodes were placed on an EEG cap (g.tec, g.GAMMAsys) according to 10-20 international system (F3, Fz, F4, T7, C3, Cz, C4, T8, P3, Pz, P4, O1, Oz, and O2) (Fig 1 (c)) and were selected to cover five main cortical regions (i.e., frontal, central, temporal, parietal and occipital) in both left and right hemispheres (red circles) and midline locations (green circles). We chose these electrodes due to their relative alignment with the brain regions that were identified by the previous research for their significant involvement in hypnosis: the default mode network (DMN) [63] and fronto-parietal network [1, 8, 10]. From a broader

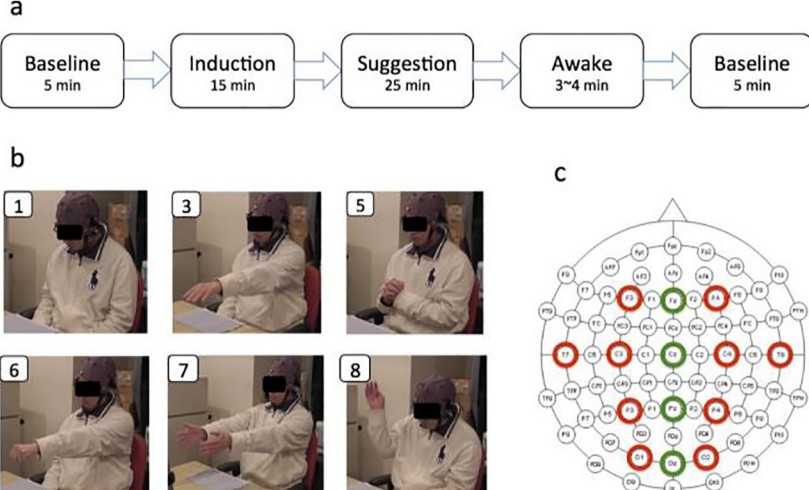

**Fig 1. Experimental procedure.** (a) The experiment was conducted in five stages. There were two baseline recordings before and after hypnosis session. Hypnosis session included induction, suggestion, and awake phases. (b) Subjects experienced hypnotic instructions that were prepared according to Harvard Group Scale of Hypnotic Susceptibility, Form A (HGSHS:A). (c) Fourteen electrodes placed on the frontal, temporal, central, parietal, and occipital areas in both left and right hemispheres (red circles) and the midline locations (i.e. green circles) recorded EEG signals during the experiment.

perspective, the channels that were included in our study covered all the major lobes of the brain that are involved in action, emotion, language, cognitive control, and action (see [64], Chapters 9 through 12 for a detailed treatment of the subject).

A reference electrode was mounted on the right ear, with a ground electrode on the forehead. Impedance of electrodes was kept below 5 kOhm by applying conductive gel. Recorded signals were amplified using g.USBamp developed at Guger Technologies (Graz, Austria). The sampling rate was 128 Hz. A 50.0 Hz notch filter was used to reduce the noise.

## Data preprocessing

We performed offline preprocessing on the recorded EEG signals, using EEGlab version 13.4.4b [65]. Data was first monitored and gross movement artefacts were excluded manually. Next, EEG time series of all channels were filtered within 0.5 to 30.0 Hz. We excluded gamma band (30-60Hz) from our analysis because the Harvard hypnosis test mainly includes motor items that require movement as a behavioural response and therefore, artefacts from muscle activity during these suggestions could have contaminated high-frequency EEG signals. Eye-movement and noises from other sources were rejected using independent component analysis (ICA) in EEGlab (eegrunica function). Then, we segmented these cleaned EEG signals into fourteen phases. They were: (1) pre-hypnosis baseline, (2) induction, (3) suggestion1, (4) suggestion2, (5) suggestion3, (6) suggestion4, (7) suggestion5, (8) suggestion6, (9) suggestion7, (10) suggestion8, (11) suggestion9, (12) suggestion10, (13) awake, (14) post-hypnosis baseline. These phases were selected based on the onset and offset of each hypnotic suggestion, as registered by the experimenter during the experiment. The rest times between the suggestions were excluded. Finally, the EEG data of each phase for every channel was decomposed into three frequency bands: theta (4-7.9 Hz), alpha (8-11.9 Hz), and beta (12-28 Hz).

## Feature extraction

We computed DE of a given frequency band (i.e., theta ($\theta$), alpha ($\alpha$), or beta ($\beta$)) for each of the fourteen EEG channels of every participant as [50]

$$H(X_j^{(f)}) = \frac{1}{2} \log_b(2\pi e \sigma_{X_j^{(f)}}^2), \ j = 1, \ldots, \ 14, \ f = \theta, \ \alpha, \ \beta \qquad (1)$$

where $\sigma_{X_j^{(f)}}^2$ is the variance of a given frequency band, $f \in [\theta, \alpha, \beta]$, in $j^{th}$ EEG channel of a participant and $H(X_j^{(f)})$ computes the entropy of the frequency band, $f$, in $j^{th}$ EEG channel of the participant. Fig 2 visualizes this feature extraction process. Although there is no restriction in selection of logarithm base, b, in Eq (1), we used b = 2, thereby interpreting the change in brain activity in bits, as originally presented by Shannon [51]. As a result, calculated DEs

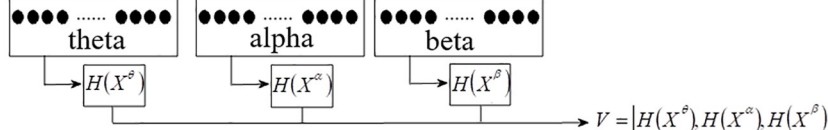

**Fig 2. Feature extraction.** The function H calculates the entropy of its input time series as per Eq (1). $X^\theta$, $X^\alpha$, and $X^\beta$ are the frequency components associated with EEG time series X. We used the resulting feature vector $V = [H(X^\theta), H(X^\alpha), H(X^\beta)]$ in our analyses.

quantified the average amount of variation in information in the brain activity in response to hypnotic suggestions.

## Statistical analyses

Given the results of Harvard test [62], we identified a total of fourteen participants in LOW (eight participants, three females, M = 25.13, SD = 6.47) and HIGH (six participants, three females, M = 24.67, SD = 5.28) suggestible groups. First, we balanced the number of participants in HIGH and LOW suggestible groups. Result of Harvard test suggested that all the LOW suggestible participants scored either one or three. Therefore, we excluded two participants with the highest score (i.e., three in our case) at random and included the remaining six LOW suggestible participants in this group. As a result, our analyses included six participants in each of LOW (three females, M = 25.83, SD = 7.48) and HIGH groups, out of which four participants (one female) had previously experienced hypnosis either in form of a stage show or a research experiment. We adapted this selection procedure from Jiang et al. [18].

Each individual experienced 5 main stages (Fig 1) that included 14 phases: a baseline recording phase and an induction (2 initial phases), 10 suggestions (10 separate phases), an awakening from hypnosis phase, and a post-baseline (2 final phases). The 10 suggestions in the middle were segments of interests in our study. We computed one DE value for each segment in each frequency band and each EEG location. Given 10 suggestions and that each of HIGH and LOW groups included 6 participants, we had $6 \times 10 = 60$ DEs, per frequency band and for each of HIGH and LOW groups (e.g., 60 DEs for alpha band at F3). In the case of FC, we used these 60 DE values, per frequency (i.e., 6 participants × 10 suggestions), per channel, to compute the pairwise correlations among the channels.

Our analyses comprised of two primary steps: 1) significance test in which we verified whether the DEs associated with the LOW and HIGH suggestible groups significantly differed 2) change in degree of synchrony in which we determined whether the observed significant difference between LOW's and HIGH's DEs was also associated with a significant change in the participants' brain regional FC in response to hypnosis suggestions. We elaborate on these steps below.

**DEs' significance test.**   For this test, we used the LOW and HIGH participants' DE values that pertained to suggestion phases (i.e., suggestion1 through suggestion10) and performed a group-wise Wilcoxon rank sum between each of the frequency bands (e.g., theta band between LOW and HIGH). Each group included 6 individuals.

**Change in functional connectivity (FC).**   To determine any potential significant change in functional connectivity among EEG channels of HIGH versus LOW suggestible groups, we performed all-pair FC analysis. For this purpose, we combined DEs of all participants for a given channel at a given frequency band and computed the pairwise FC using Pearson correlation (i.e., every pair of channels). This resulted in $14 \times 14$ FC matrices, per frequency band, where 14 refers to the number of EEG channels. For each channel, we then computed the average Pearson correlations that it had with the remainder of the channels and only considered those channels whose averaged Pearson correlations were $\geq 0.70$ (i.e., primarily strong and very strong correlations) in our analysis. For the selected channels, we also counted the number of channels that they were synchronized with (i.e., number of channels that they showed $\geq 0.70$ correlation with). For both of these measures (i.e., averaged correlation and number of synchronized channels, per selected channel), we used Kruskal-Wallis test to determine the effect of suggestibility on FC. We followed this test with post hoc paired Wilcoxon rank-sum test.

For Kruskal-Wallis, we reported the effect size $r = \sqrt{\frac{\chi^2}{N}}$, as suggested by Rosenthal and DiMatteo [66]. In the case of Wilcoxon test, we used $r = \frac{W}{\sqrt{N}}$ [67] as effect size with $W$ denoting the Wilcoxon statistics. $N$ is the sample size in both cases. The effect size in non-parametric tests is considered [68] small when r $\leq$ 0.3, medium when $0.3 <$ r $< 0.5$ and large when r $\geq$ 0.5.

### Ethics statement

All subjects singed a written informed consent from in accordance with ethical approval of the Ethics Committee (Approval number: 412-2), University of Tokyo. Every participant received a payment at the end of the experiment.

## Results

### DEs' significance test

Fig 3 shows the results of Wilcoxon rank sum on frequencies (i.e., theta, alpha, and beta) of fourteen EEG channels of HIGH and LOW suggestible participants. Although we observed significant differences between paired frequencies (e.g., theta band in HIGH and LOW groups) in all EEG channels (shown in Table 1, column p <), their differences exhibited a varying degree of effect (Table 1, column r). In what follows, we highlight the brain regions that exhibited strong effect sizes (i.e., r $\geq$ 0.50) in two or more frequency bands. Table 1 provides the full results of the significant differences, per brain region, per frequency.

In the frontal region, we observed a large effect size between theta as well as alpha of HIGH and LOW suggestible groups at all channels as shown in the first row of Fig 3, namely, F3, Fz, and F4.

In the case of temporal regions (Fig 3, left and right subplots) we observed large-effect significant difference in theta and alpha bands at both T7 and T8 locations and a large-effect significant difference in beta band only at T8.

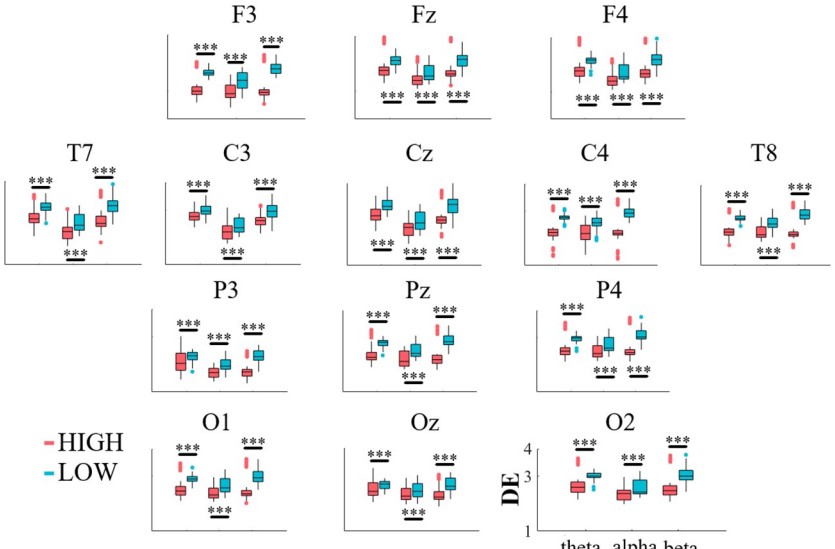

**Fig 3. Descriptive statistics of theta, alpha, and beta frequencies of EEG time series of HIGH and LOW suggestible groups.** Subplots are organized based on EEG 10-20 system. Y-axis represents the differential entropy (DE) of these frequency bands. This axis is within 1-4 (bits, given b = 2 in Eq 1) range in all subplots. Asterisks mark the significant difference between the corresponding frequencies in HIGH and LOW suggestible groups (***: p <.001).

**Table 1. Paired Wilcoxon rank sum between LOW and HIGH suggestible subjects.** $r = \frac{W}{\sqrt{N}}$ [67] is the effect size with N and W representing the sample size and the Wilcoxon statistics, respectively. $M_L$, $SD_L$, $M_H$, and $SD_H$ are the mean and standard deviation of the DE values of a given frequency for LOW (i.e., subscript L) and HIGH (i.e., subscript H) suggestible groups.

| | p < | W(118) | r | $M_L$ | $SD_L$ | $M_H$ | $SD_H$ |
|---|---|---|---|---|---|---|---|
| $\theta_{F3}$ | .001 | 6.39 | .59 | 2.80 | .19 | 2.11 | .49 |
| $\alpha_{F3}$ | .001 | 6.47 | .60 | 2.60 | .15 | 2.12 | .40 |
| $\beta_{F3}$ | .001 | 5.27 | .49 | 2.36 | .32 | 1.98 | .34 |
| $\theta_{Fz}$ | .001 | 5.83 | .54 | 3.21 | .31 | 2.85 | .52 |
| $\alpha_{Fz}$ | .001 | 5.70 | .52 | 3.16 | .17 | 2.89 | .50 |
| $\beta_{Fz}$ | .001 | 4.16 | .38 | 2.70 | .30 | 2.47 | .31 |
| $\theta_{F4}$ | .001 | 5.74 | .53 | 3.14 | .29 | 2.81 | .52 |
| $\alpha_{F4}$ | .001 | 5.61 | .52 | 3.10 | .19 | 2.88 | .48 |
| $\beta_{F4}$ | .001 | 3.35 | .31 | 2.65 | .28 | 2.49 | .32 |
| $\theta_{T7}$ | .001 | 6.35 | .58 | 3.14 | .34 | 2.64 | .44 |
| $\alpha_{T7}$ | .001 | 6.04 | .56 | 3.10 | .21 | 2.74 | .35 |
| $\beta_{T7}$ | .001 | 4.48 | .41 | 2.50 | .30 | 2.21 | .40 |
| $\theta_{C3}$ | .001 | 5.65 | .52 | 2.96 | .32 | 2.63 | .26 |
| $\alpha_{C3}$ | .001 | 4.62 | .42 | 2.30 | .24 | 2.79 | .23 |
| $\beta_{C3}$ | .001 | 2.76 | .25 | 2.41 | .28 | 2.26 | .36 |
| $\theta_{Cz}$ | .001 | 6.14 | .57 | 3.24 | .39 | 2.77 | .42 |
| $\alpha_{Cz}$ | .001 | 5.79 | .53 | 3.24 | .27 | 2.88 | .34 |
| $\beta_{Cz}$ | .001 | 3.82 | .35 | 2.61 | .34 | 2.36 | .33 |
| $\theta_{C4}$ | .001 | 6.38 | .59 | 2.93 | .20 | 2.25 | .57 |
| $\alpha_{C4}$ | .001 | 6.37 | .59 | 2.78 | .13 | 2.26 | .47 |
| $\beta_{C4}$ | .001 | 4.93 | .45 | 2.56 | .28 | 2.17 | .42 |
| $\theta_{T8}$ | .001 | 6.45 | .59 | 2.90 | .22 | 2.28 | .47 |
| $\alpha_{T8}$ | .001 | 6.36 | .59 | 2.77 | .13 | 2.31 | .35 |
| $\beta_{T8}$ | .001 | 5.93 | .55 | 2.59 | .27 | 2.25 | .26 |
| $\theta_{P3}$ | .001 | 7.34 | .67 | 2.30 | .22 | 1.77 | .31 |
| $\alpha_{P3}$ | .001 | 3.68 | .34 | 2.07 | .38 | 2.29 | .17 |
| $\beta_{P3}$ | .001 | 5.47 | .50 | 1.99 | .27 | 2.71 | .20 |
| $\theta_{Pz}$ | .001 | 6.52 | .60 | 2.89 | .22 | 2.33 | .45 |
| $\alpha_{Pz}$ | .001 | 5.99 | .55 | 2.80 | .13 | 2.40 | .38 |
| $\beta_{Pz}$ | .001 | 5.05 | .47 | 2.52 | .27 | 2.20 | .28 |
| $\theta_{P4}$ | .001 | 6.35 | .58 | 3.08 | .23 | 2.61 | .46 |
| $\alpha_{P4}$ | .001 | 6.15 | .57 | 2.96 | .13 | 2.62 | .39 |
| $\beta_{P4}$ | .001 | 4.40 | .41 | 2.72 | .27 | 2.50 | .27 |
| $\theta_{O1}$ | .001 | 6.35 | .58 | 2.98 | .24 | 2.51 | .45 |
| $\alpha_{O1}$ | .001 | 6.11 | .56 | 2.90 | .14 | 2.57 | .38 |
| $\beta_{O1}$ | .001 | 4.79 | .44 | 2.65 | .27 | 2.40 | .26 |
| $\theta_{Oz}$ | .001 | 5.91 | .54 | 2.67 | .25 | 2.32 | .33 |
| $\alpha_{Oz}$ | .001 | 3.60 | .33 | 2.66 | .16 | 2.49 | .29 |
| $\beta_{Oz}$ | .001 | 2.59 | .24 | 2.45 | .27 | 2.33 | .25 |
| $\theta_{O2}$ | .001 | 5.94 | .55 | 3.04 | .30 | 2.63 | .46 |
| $\alpha_{O2}$ | .001 | 5.82 | .54 | 2.30 | .15 | 2.70 | .42 |
| $\beta_{O2}$ | .001 | 3.13 | .29 | 2.56 | .28 | 2.37 | .26 |

In the central regions (Fig 3, middle subplots), we observed such large-effect significant differences in theta and alpha bands at Cz and C4 only.

In parietal area, shown in third row of Fig 3, such significant differences with large effect sizes were observed in theta and beta bands at P3 along with theta and alpha bands at Pz, P4.

In the occipital region (fourth row of Fig 3), we observed large-effect significant differences between theta as well as beta bands of HIGH and LOW suggestible groups at O1 and O2.

Taken together, our analyses indicated that the significant differences that were charactrized with a large effect size were mainly associated with theta and alpha bands. In the case of beta band, such differences were primarily observed at T8 with only a marginally large effect size at P3.

## Change in functional connectivity (FC)

Fig 4 depicts the functional connectivity density in HIGH and LOW suggestible groups, per frequency band. Between-group Kruskal-Wallis test identified a significant effect of suggestibility (p <.001, H(1, 83) = 23.21, $\eta^2$ = .28). Post hoc analysis of this result (Table 2) indicated a significantly higher functional connectivity in HIGH suggestible groups with respect to theta and alpha bands. On the other hand, it indicated (Table 2) a non-significant difference in the beta band.

Fig 5 shows the paired connectivity map of the channels in HIGH and LOW suggestible groups. Between-group Kruskal-Wallis test indicated the significant effect of suggestibility on number of channels that different channels were synchronized with (i.e., number of channels that they showed $\geq$.70 correlation with) (p <.01, H(1, 83) = 6.84, $\eta^2$ = .08). Post hoc analysis of this result (Table 3) implied a significantly higher number of synchronized channels in HIGH compared to LOW suggestible groups in the case of theta (Fig 5(a)) and alpha (Fig 5(b)) and a non-significantly lower number of synchronized channels in HIGH versus LOW in beta band (Fig 5(c)).

## Discussion

In this article, we examined the utility of DE as a reliable biomarker for quantification of the brain neural responses to hypnotic suggestibility. In doing so, we attributed the inconsistencies among the findings in hypnosis literature [33, 34] to application of the overall change in power (e.g., averaging the change in power amplitude in a given frequency band) in their analyses which ignored the crucial role of the brain variability in its functioning [45, 46]. Our approach was motivated by the viewpoint that advocates the possibility of the presence of significant information in the absence of any observable elevation in power [38].

Fano factor (i.e., $F = \frac{\sigma^2}{\mu}$) [61] characterizes the neural spiking as a deviation of activation of neural population from their expected firing rate. It is apparent that such a deviation is minimized when responses of a given neural population is in unison in which case the firing of every individual neuron tends to the expected firing rate of the entire neural population (i.e., $\sigma^2 \rightarrow 0$). In this respect, it appears plausible to construe the observed lower information content (and hence the variability) in the HIGH suggestible participants' theta, alpha, and beta frequency bands as a marker of a neural population that exhibits a highly (i.e., in comparison with LOW suggestible participants) synchronized activity to hypnotic suggestions. This interpretation is in accord with Keshmiri et al. [60] on direct correspondence between the change in variation in information and neural spiking rate. It also finds further support in recent findings by Wittig Jr. et al. [69] that showed that the spiking neuronal activity was suppressed and became more reliable in preparation for verbal memory formation. In the present study, this

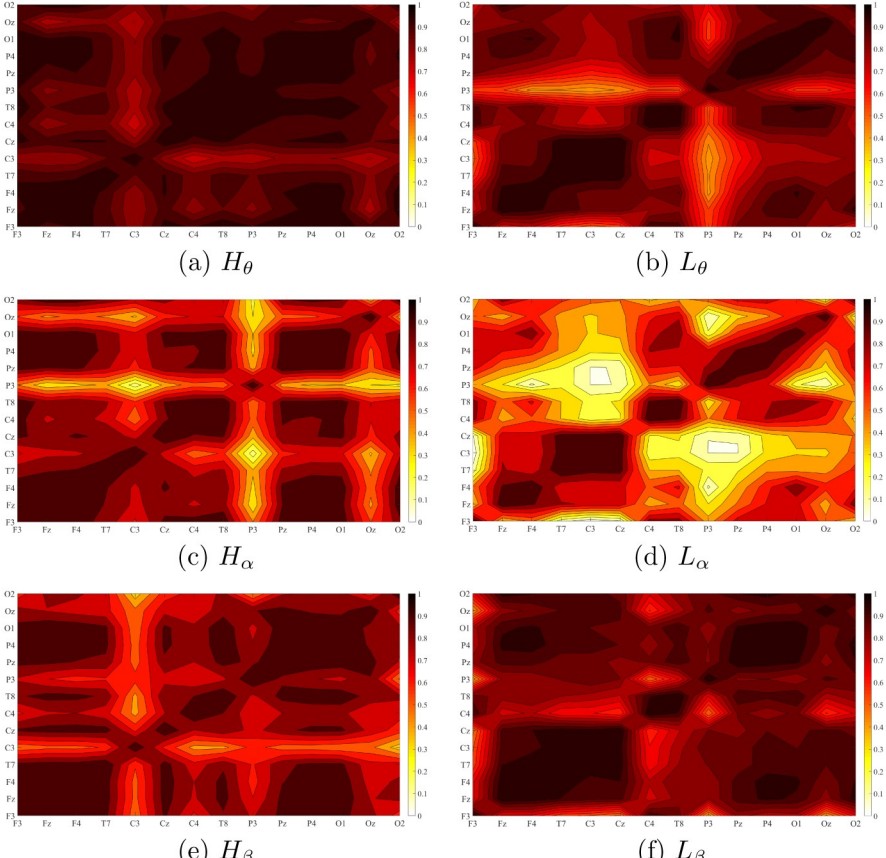

**Fig 4. Grand-average FC of the participants using paired Pearson correlation between EEG channels in HIGH (left column: (a) theta (c) alpha (e) beta) and LOW (right column: (b) theta (d) alpha (f) beta) suggestible participants.** In case of HIGH suggestible group, we observed significantly higher FC in theta and alpha bands, mostly in left parietal (P3, theta and alpha) as well as occipital (Oz, alpha). On the other hand, we observed a non-significantly lower FC in the case of beta band, approximately around the left central (C3) region. These subplots identify an overall increase in strength of FC in theta and alpha bands that is accompanied by an overall weakening in beta FC in case of HIGH compared to LOW suggestible groups. (a) $H_\theta$ (b) $L_\theta$ (c) $H_\alpha$ (d) $L_\alpha$ (e) $H_\beta$ (f) $L_\beta$.

interpretation was also evident in the significantly higher FC in the theta and alpha frequency bands in the case of HIGH compared to LOW suggestible participants.

We observed that the large effect of hypnotic suggestibility on information content of the theta, alpha, and beta frequency bands was not confined to the EEG channels that covered a specific hemisphere but manifested (with comparable strength in their effect sizes) in both,

**Table 2. Wilcoxon rank sum test of FC between LOW and HIGH suggestible subjects.** $r = \frac{W}{\sqrt{N}}$ [67] is the effect size with N and W representing the sample size and the Wilcoxon statistics, respectively. $M_L$, $SD_L$, $M_H$, and $SD_H$ are the mean and standard deviations of the given frequency for LOW (i.e., subscript L) and HIGH (i.e., subscript H) suggestible groups.

|  | **p <** | **W(26)** | **r** | **$M_L$** | **$SD_L$** | **$M_H$** | **$SD_H$** |
|---|---|---|---|---|---|---|---|
| $\theta$ | 001 | 3.93 | .77 | .91 | .04 | .82 | .07 |
| $\alpha$ | .001 | 3.51 | .69 | .78 | .13 | .58 | .08 |
| $\beta$ | = .19 | -1.31 | .26 | .82 | .09 | .85 | .06 |

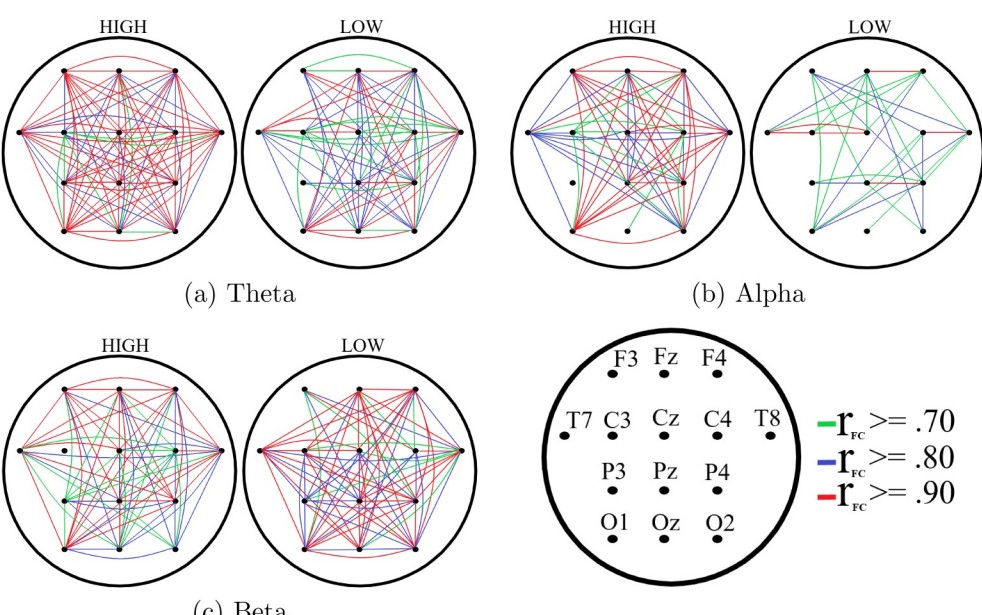

**Fig 5. Grand averages of the change in FC among EEG channels of HIGH and LOW suggestible subjects: (a) theta (b) alpha (c) beta frequency bands.** In these subplots, the left map is associated with HIGH and the right map corresponds to LOW groups. In case of HIGH suggestible group, we observed higher regional connection counts in theta and alpha bands along with a lower connection counts in their beta band. Substantially higher connectivity in case of HIGH suggestible group in theta and alpha bands is evident in these subplots. $r_{FC}$ refers to Pearson correlation coefficient based on which FC among channels was determined. (a) Theta (b) Alpha (c) Beta.

EEG channels on the left as well as the right hemispheres. This, in turn, suggested that the brain activity of HIGH suggestible participants (i.e., in comparison with LOW participants) exhibited a rather global neural responses to hypnotic suggestions whose effect was significantly distributed between their left and right hemispheric neural activity. Recent study by Han et al. [70] on projection patterns of 591 individual neurons in the mouse primary visual cortex revealed that most neurons targeted multiple cortical areas, often in non-random combinations. Furthermore, their results indicated that the signals that were carried by individual cortical neurons were shared across subsets of target areas, and thus concurrently contributed to multiple functional pathways. These findings provided a promising evidence in support of the global brain neural excitation in response to stimuli, as observed in our results in the case of HIGH suggestible participants during the hypnotic suggestions.

Our results also identified a significantly higher FC in HIGH suggestible participants' theta and alpha bands that was more pronounced in the parietal (in the case of theta) and centroparietal (for alpha) regions and that was accompanied by a non-significantly smaller FC in the

**Table 3. Wilcoxon rank sum of the connectivity maps between LOW and HIGH suggestible subjects.** $r = \frac{W}{\sqrt{N}}$ [67] is the effect size with N and W representing the sample size and the Wilcoxon statistics, respectively. $M_L$, $SD_L$, $M_H$, and $SD_H$ are the mean and standard deviations of the given frequency for LOW (i.e., subscript L) and HIGH (i.e., subscript H) suggestible groups.

|          | p <   | W(26)  | r   | $M_L$ | $SD_L$ | $M_H$ | $SD_H$ |
|----------|-------|--------|-----|-------|--------|-------|--------|
| $\theta$ | .001  | 4.55   | .89 | 13.00 | 0.0    | 10.43 | 2.77   |
| $\alpha$ | .01   | 3.19   | .63 | 9.00  | 3.76   | 4.71  | 1.90   |
| $\beta$  | = .20 | -1.29  | .25 | 10.14 | 3.16   | 11.14 | 2.21   |

beta band in central region. In this respect, Jamieson and Burgess [39] also reported similar changes in FC from pre-hypnosis to hypnosis state using iCOH (increase in theta) [43] and COH (decrease in beta1) [42]. However, their analyses which were primarily based on the state of hypnosis (i.e., without observing responses of the participants to hypnotic suggestions) did not identify any significant differences [39] between pre-hypnosis and hypnosis state on these bands. Our results complemented these findings by extending the observed effects from neutral hypnosis to hypnotic suggestions, thereby identifying their presence as a potential signature of hypnotic state. Moreover, our findings improved their results by introducing a single measure (i.e., DE in contrast to iCOH and COH for the theta and beta bands, respectively) that was able to capture the significant differences between HIGH and LOW suggestible participants.

Burgess and Gruzelier [71] suggested the potential role of alpha oscillations for a hippocampally dependent large-scale integration of information across brain areas that were distributed over temporal, fronto-parietal, and occipital regions. A number of previous findings also proposed the role of theta band in transfer of information between the hippocampus and the neocortex [72, 73] as well as in reflecting the intensification of attentional processes [74]. Our findings on the effectiveness of DE as an information-theoretic measure of brain neural responses to hypnosis suggestions were in line with these findings on the role of alpha and theta bands in information processing and transfer of information between functionally connected brain regions. Additionally, a comprehensive review by Perlini and Spanos [33] on the contribution of alpha band to hypnosis responses of human subjects concluded that the observed tentatively positive findings on the role of alpha band in the hypnotic state required further investigation to ensure the reproducibility of its effect. In this respect, previous findings during neutral hypnosis [14, 41] on the significant increase in alpha FC in conjunction with our results on differentially large effect of alpha during hypnotic suggestions provided further evidence for the substantial role of this frequency band during hypnosis.

Another interesting observation was the apparent higher frontal area's FC in the HIGH compared to LOW suggestible groups. Rainville and Price [1] showed that the absorption-related effect included increased activation in the frontal and posterior parietal regions. Moreover, Bell et al. [75] identified that an increase in prefrontal cortex activity indicated the potential involvement of the executive system during hypnosis suggestions that was accompanied by an increased occipital regional blood flow (rCBF) [15]. They also showed that this increase in occipital rCBF was negatively correlated with hypnotic absorption [2]. These findings pointed at the engagement of executive attentional network [1] during hypnotic experience. This view found further evidence in the requirement of the attentional processes for selective enhancement of target-stimulus processing as well as inhibition of competing processes and responses [18, 76–78]. In this respect, the ability of DE for quantification of the enhanced frontal activity in HIGH suggestible participants along with the observed increases in the occipital channels contributed to these findings and their interpretation of hypnosis as an altered state of consciousness [1, 2].

We also observed a higher FC between temporal and occipital channels in the case of HIGH suggestible participants. Previous research also identified the occurrence of such temporo-occipital functional linkings in response to visual stimulation [79, 80]. Although it is plausible to attribute this to the open-eye effect during hypnosis session, it is an unlikely expectation in our case since our subjects had their eyes close throughout the experiment (except for the pre- and post-baselines which are not included in our analyses). Therefore, it is possible to propose that this effect, that was also reported by Fingerkurts and colleagues [14] during the neutral hypnosis of a single hypnosis virtuoso (eyes open in their case), is a neuro-cognitive marker of hypnosis. In fact, Fingerkurts and colleagues [14] suggested that this effect might

indicate the participants' readiness processing of the suggestions and translation to hallucinated realities in perception. However, this hypothesis requires further investigation to test for its validity.

A number of mental and behavioural disorders are charactrized by peculiar functioning of the brain neural activity that are observable in the theta and alpha frequencies [23–26]. Moreover, the use of hypnotic suggestions to suppress episodic memories (post-hypnotic amnesia) implies alterations in the brain areas responsible for long-term memory retrieval (i.e., occipital, temporal, and prefrontal) [81, 82]. Our results on ability of DE in capturing the significant effect of hypnotic suggestions on these frequencies along with its utility in quantification of the observed distributed brain activity in response to hypnotic suggestions (with its effect most pronounced in frontal, temporal, and occipital regions) hint at its utility as a robust biomarker for study of the effect of hypnosis suggestions on the brain. They further highlight the potential of DE as an adequate biomarker for quantification of the effect of hypnosis on brain neural responses during the treatment of such behavioural and mental disorders.

## Limitations and future direction

Our results identified DE as a potential unifying measure to reproduce previous observations that were based on multiple biomarkers. DE also appeared to further complement these measures by extending their results during the neutral hypnosis to the case of hypnotic suggestions, thereby identifying such neural activations as potential signatures of hypnotic experience. However, our results did not compare DE with these previous measures (e.g., COH). Therefore, future comparative analyses of DE and these other measures to clarify their respective dis/advantages will be necessary to thoroughly appreciate their proper domain of application.

We also primarily focused on determining whether DE can identify the subtle differences between high versus low suggestible individuals' neural responses to hypnotic suggestions. Therefore, we did not include the individuals' pre-hypnosis rest period. Inclusion of such baselines (e.g., as control signals) can help determine whether DE can also quantify the change in the brain activity of these individuals from their respective pre-hypnosis rest time. This is indeed an important and interesting venue for future research, considering the association between hypnotizability and the brain activity during attention outside hypnosis [10] and the potential role of such a responsiveness in prediction of the individuals' suggestibility [13].

Given our primary objective, we inevitably discarded a rather larger sample of individuals that were categorized as mid-hypnotizable group. However, inclusion of these individuals whose responses to hypnotic suggestions are not distinctively low or high, can potentially shed light on the nature of the observed variation of information in the brain during hypnosis. For instance, DE might be useful for determining whether the change from low to high suggestibility occurs along a continuum that encompasses the mid-suggestible group's brain activity or such differences are rather associated with distinct and mutually exclusive neural responses.

Considering the crucial role of the cortical self-organized criticality [83–85] in maximizing its information capacity [86–88], entropy has been proven as a powerful tool for quantification of the variability in brain functioning [89] and cortical activity [90] in such broad area of research as information processing capacity of working memory (WM) [47] and the state of consciousness [91]. Although the use of DE in neuroimaging (e.g., Tononi et al. [92] and Carhart-Harris [89]) and EEG studies (e.g., Duan et al. [56], Zheng and Lu [57], Shi et al. [58], and Zheng et al. [93]) has presented promising results, its application for modeling of the brain functioning requires further investigation. Specifically, parametric adaptation of DE for the analysis of EEG time series [56–59, 93] assumes that such data is homoscedastic and normally distributed. While the applicability of such an assumption in neuroimaging studies has been

investigated [60, 94, 95], similar theoretical studies to better position the use of DE in EEG-based brain research is currently (to the best of our knowledge) lacking. Such analyses can help determine the domain of applications in which DE may not be an adequate measure for modeling the EEG time series of the brain activity. Along the same direction, it is also interesting to further examine the utility of the non-parametric formulation of the differential entropy [96, 97] for modeling of EEG time series [98].

In the present study, we were specifically interested in behavioural responses to hypnotic suggestions that were mainly ideo-motor suggestions, inducing movement and therefore noise in EEG. Therefore, we decided to exclude the gamma band from our analyses, considering its vulnerability to movement-related artefacts. Subsequently, we opted for a lower sampling rate of 128 Hz for EEG recordings. This choice was in accord with our overview of the EEG-based hypnosis research that identified 128 Hz and 256 Hz as the most commonly used sampling rates [14, 31, 99]. However, future research that is empowered with high density electrodes and that includes higher frequency bands can allow for more comprehensive realization of the underlying dynamics of brain responses to hypnotic suggestions. Such setting can also provide better testbeds for critical examination of DE and other biomarkers for study of hypnosis.

In spite of the fact that the original group of individuals who participated in our hypnosis experiment formed a moderately acceptable sample size (i.e., forty-six subjects), the final validation for their inclusion in LOW and HIGH groups based on Harvard test [62] resulted in a small sample. Furthermore, all of these individuals were university students/staff, some of whom had previous exposure to hypnosis experience. As a result, it is plausible to presume that our participants were able to more readily comprehend and follow our experimental procedure, thereby contributing to an above-average outcome that one might expect from a general population. Therefore, it is crucial to reevaluate these findings while considering a broader general population.

The present findings are not only of interest to the psychology and neuroscience community but also to the researchers in the field of AI and brain-computer interfaces [100, 101]. For instance, the use of DE as a biomarker of hypnosis can be utilized in development of real-time EEG classifiers that detect their users' responses to hypnotic suggestions. This, in turn, can expedite the deployment of the automated hypnotherapeutic systems of the future for clinical treatment of mental and behavioural disorders at brain functional level [23–26]. Such adaptations, in turn, can take the field a step closer to personalized hypnosis interventions that are tailored around the individuals' suggestibility level.

## Supporting information

**S1 Data.**
(ZIP)

## Author Contributions

**Conceptualization:** Maryam Alimardani, Kazuo Hiraki.

**Data curation:** Maryam Alimardani, Kazuo Hiraki.

**Formal analysis:** Soheil Keshmiri.

**Funding acquisition:** Soheil Keshmiri, Maryam Alimardani, Masahiro Shiomi, Kazuo Hiraki.

**Methodology:** Maryam Alimardani, Kazuo Hiraki.

**Project administration:** Kazuo Hiraki.

**Resources:** Kazuo Hiraki.

**Supervision:** Masahiro Shiomi, Hidenobu Sumioka, Hiroshi Ishiguro, Kazuo Hiraki.

**Validation:** Kazuo Hiraki.

**Writing – original draft:** Soheil Keshmiri.

**Writing – review & editing:** Soheil Keshmiri, Maryam Alimardani.

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
