## [Decision Letter · Decision Letter 0]

12 Feb 2020

PONE-D-20-00814

Higher Hypnotic Suggestibility Is Associated with the Lower EEG Signal Variability in Theta, Alpha, and Beta Frequency Bands

PLOS ONE

Dear Dr. Keshmiri,

Thank you for submitting your manuscript to PLOS ONE. After careful consideration, we feel that it has merit but does not fully meet PLOS ONE’s publication criteria as it currently stands. Therefore, we invite you to submit a revised version of the manuscript that addresses the points raised during the review process.

We would appreciate receiving your revised manuscript by Mar 28 2020 11:59PM. To enhance the reproducibility of your results, we recommend that if applicable you deposit your laboratory protocols in protocols.io, where a protocol can be assigned its own identifier (DOI) such that it can be cited independently in the future. For instructions see: http://journals.plos.org/plosone/s/submission-guidelines#loc-laboratory-protocols

We look forward to receiving your revised manuscript.

Kind regards,

Vilfredo De Pascalis

Academic Editor

PLOS ONE

Journal Requirements:

2. Please provide additional details regarding participant consent. In the ethics statement in the Methods and online submission information, please ensure that you have specified whether consent was informed.

Reviewers' comments:

Reviewer's Responses to Questions

**Comments to the Author**

1. Is the manuscript technically sound, and do the data support the conclusions?

Reviewer #1: Yes

Reviewer #2: Yes

2. Has the statistical analysis been performed appropriately and rigorously? 

Reviewer #1: Yes

Reviewer #2: Yes

3. Have the authors made all data underlying the findings in their manuscript fully available?

Reviewer #1: Yes

Reviewer #2: Yes

4. Is the manuscript presented in an intelligible fashion and written in standard English?

Reviewer #1: Yes

Reviewer #2: Yes

5. Review Comments to the Author

Reviewer #1: General Overview:

This study investigates and highlights the search for an effective and robust biomarker in the EEG to delineate hypnotisability. Overall this is a well written and researched article with solid justification for the use of an information content approach.

The study concept is certainly worthy of publication however, will require some minor additional amendments both in grammar and with some minor additional proof reading with respect to processes and assumptions. Also, the limitations of the study should be addressed in the final discussion; role of high density electrodes approaches, sampling sizes stability of hypnotized state, potential for artifactual findings, false positives etc.

Overall, this is an innovative study, which was conceptualised, designed and tested well and will certainly add to the literature and field.

Abstract

The clarity of abstract’s argument and discussion of findings, may need some proof reading. eg the sentence with “summary statistics” unclear. This abstract doesn’t include details of the population tested and how. Refer to last paragraph in Introduction comment below. Maybe include this instead.

Introduction

• Very well written and latest research discussed well. Good Structured narrative.

• Justification for this direction of research argued and presented well.

• P2 line 46. The use of the word “recently” to reflect work that is 13 yrs old. [ref 14]

• Line 58 COH beta effect also seen in other studies (eg Deivanayagi et al 2007) and some did not (Sabourin et al 1990 ref 30). May need to clarify this in line 58. 68 why? There are some consistencies.

• Line 97 Why discuss the methodology/results and conclusions of the experiment in the introduction? Should only be discussing the justification of the study. Some of this material would be better in the abstract.

Material and Methods

• Line 127. Would a broader community sample have been more useful than a possibly biased academic one?

• Line 146: Would it also have been on some research interest to also examine the EEG of the mid hypnotisability group, to see if there is a linear relationship on response differences? This may have been useful.

• What were the gender distribution of the remaining 14 (8L, 6H) tested? How many of these had previously been hypnotized?

• Line 151.” placed the electrodes.” How, by what scheme; electro cap should be explained, (ref?) etc?

• Fig 1: were the other electrodes recorded or were just 14 recorded? If so, why were these selected? The electrode map may be misleading if the other electrodes were not used.

• Line 176. A faster sampling rate may have been less problematic (especially for measuring higher EEG frequencies).

• Line 182. It would have been difficult to record gamma anyway with the low sampling rate.

• Line 192. Why were the rest times excluded? These could have been associated with a control baseline.

• Line 208. How was this balanced? 8 vs 6? Or another process?

• Line 222. How were 60 DE values calculated, based on time? Why 60? Needs clarifying.

Results

• Line 284 Should read “Change in Functional Connectivity (FC)”

• Fig 5: Clarify that the left COH map is High and right is low.

• It would have been interesting to see a comparison with standard coherence maps for each band with the changes in FC.

• Requires some clarity about how the 14 states were used to produce the data for subsequent analyses (average of suggestion1-10?). This is mentioned in Discussion line 349. Should be addressed in Results in more detail and why?

Discussion

• A well written discussion about the effects/implications for the findings, however there should also be a discussion about the potential limitations of the study.

• There should also be some discussion about how the study could be expanded in future.

• When is DE not an adequate biomarker? what are the bare minimum requirements to be able to calculate it? Eg Can you do a case study on an individual? Perhaps this should be discussed in Methods as well. By citing and including the work of Duan et al 2013, Wei et al 2020, Lu et al 2020(changes in emotions), Wang et al 2020 (cognitive Control), may be useful.

References:

• Might be useful to add other studies cited above if used in narrative.

Reviewer #2: Page 1 (Abstract)

Change “Variational information” to “Variation of information”

Change “accompanies with a” to “accompanied by a”.

Change “provides a direct” to “provides direct”.

Page 2 lines 34 and 35

The phrase “power and information do not necessarily need to be modulated” is confusing in the context of the wider sentence. I suspect the authors intend to convey either that modulation of power does not necessarily involve modulation of information or that modulation of information does not necessarily involve change in local power or BOTH. The sentence should be rewritten to better convey the authors intended meaning.

Page 3 line 68 “significant differences” should probably be changed to read “significant differences in power”. Line 73 change “perspective” to “perspectives” and change “as role” to “as the role”.

Page 6 line 180 “major artefacts” is simply too vague and in-descriptive consider changing this to something like “gross movement artefacts”.

Results

Page 8 figure 3 legend P<.05 and p<.01 are un-necessary as there are no such results displayed.

Numerical values presented in Table 1 should not be reproduced in text on page 8 and page 10. The values shown in Table 1 appear to be inconsistent with the corresponding images displayed in Figure 3 for the means for lows and highs for beta at P3 and also for alpha at O2. Please check this. Consider if Figure 3 adds value for the reader in understanding these results.

Page 11 line 287 presents an eta squared value as does line 296. Please check is eta squared the intended/appropriate/correct effect size measure?

Repetition of numerical values from table 2 and table 3 in text on page 11 (and page 12) appears redundant and un-necessary. This is probably best removed.

Discussion

It may be worth acknowledging that dendritic potentials drive scalp recorded EEG alongside axonal spiking.

Page 12 Line 329 refers to findings in separate hemispheres. This seems to imply hemisphere specific tests were conducted and reported but I do not recall seeing such. Please clarify this issue for the readers.

Page 13. Lines 348 – 349 “did not identify any significant differences” did you mean “differences in alpha”? If so please state that or otherwise clarify the frequency bands referred to.Line 370 change “the hypnosis” to “hypnosis”.

Page 14 line 392 change “to speculate” to read “to propose that”.

Line 402 reference [77] consider adding an additional recent relevant finding on the role of functional connectivity in upper alpha in hypnotic amnesia suggestion responses

Jamieson, G. A., Kittenis, M. D., Tivadar, R. I., & Evans, I. D. (2017). Inhibition of retrieval in hypnotic amnesia: dissociation by upper-alpha gating. Neuroscience of consciousness, 3(1).

Note my conflict of interest I am a co-author. Please do not include this reference here unless you consider it adds value for the reader.

6. PLOS authors have the option to publish the peer review history of their article (what does this mean?). If published, this will include your full peer review and any attached files.

Reviewer #1: No

Reviewer #2: No

---

## [Author Response · Author response to Decision Letter 0]

21 Feb 2020

First and foremost, the authors would like to express their gratitudes for the reviewers’ time and kind consideration to review their manuscript. The comments by the reviewers certainly helped improve the quality of the results as well as their presentations instructively and substantially.

In the pages that follow, we provide our point-by-point responses to the reviewers’ comments.

Sincerely,

Responses to Reviewer 1

Abstract

Reviewer’s Comment: The clarity of abstract’s argument and discussion of findings, may need some proof reading. eg the sentence with “summary statistics” unclear.

Authors’ Response: We replaced “summary statistics” with the term biomarker. For further changes to the Abstract, please see our response to reviewer’s comment “This abstract doesn’t include details...”

Reviewer’s Comment: This abstract doesn’t include details of the population tested and how. Refer to last paragraph in Introduction comment below. Maybe include this instead.

Authors’ Response: To provide brief information about participants and the frequencies used in this study, we added the following to the Abstract.

“For this purpose, we utilized differential entropy (DE, i.e., the average information content in a continuous time series) of theta, alpha, and beta frequency bands of fourteen-channel EEG time series recordings that pertain to the brain neural responses of twelve carefully selected high and low hypnotically suggestible individuals.”

With regards to the changes applied to the last paragraph in Introduction, please refer to our response to reviewer’s comment “Line 97 Why discuss the methodology/results and conclusions...”

Introduction

Reviewer’s Comment: P2 line 46. The use of the word “recently” to reflect work that is 13 yrs old. [ref 14]

Authors’ Response: In the revised version of the manuscript, “recently” is removed.

Reviewer’s Comment: 68 why? There are some consistencies.

Authors’ Response: The authors are not certain whether they follow the reviewer’s comment and will be thankful if the reviewer kindly provides further information.

Reviewer’s Comment: Line 58 COH beta effect also seen in other studies (eg Deivanayagi et al 2007) and some did not (Sabourin et al 1990 ref 30). May need to clarify this in line 58.

Authors’ Response: To clarify this point, we added the following to the Section Introduction, lines 60-68, in the current version of the manuscript.

“For instance, Deivanayagi et al. [28] found that COH associated the state of hypnosis with lowered theta and alpha frequency bands. They further envisioned the use of this measure to study the effect of hypnosis on higher frequencies such as beta and gamma bands. In contrast, Sabourin et al. [31] found that COH indicated an increase in theta power during hypnosis in both low as well as high hypnotizable individuals. They further observed that the change in alpha power was not a predictor of hypnotic susceptibility, that highly susceptible subjects had more beta activity in the left than right hemispheres, and that low susceptible subjects showed only a weak lateralized asymmetry.”

Reviewer’s Comment: Line 97 Why discuss the methodology/results and conclusions of the experiment in the introduction? Should only be discussing the justification of the study. Some of this material would be better in the abstract.

Authors’ Response: We addressed this issue in two steps.

1. Abstract: We added the following information to the Abstract of the manuscript.

“For this purpose, we utilized differential entropy (DE, i.e., the average information content in a continuous time series) of theta, alpha, and beta frequency bands of fourteen-channel EEG time series recordings that pertain to the brain neural responses of fourteen carefully selected high and low hypnotically suggestible individuals.”

2. Introduction: We modified the content of this part (Section Introduction, lines 108-117, in the current version of the manuscript) as follows:

“Given these findings, we sought the utility of DE for quantification of the brain neural responses to hypnotic suggestions. Specifically, we utilized DE of the theta, alpha, and beta frequency bands of fourteen-channel EEG recordings of twelve carefully selected high and low hypnotically suggestible individuals. We found that the higher hypnotic suggestibility was associated with a significantly lower variability in information content of theta, alpha, and beta frequencies. We also observed that such a lower variability was accompanied by a significantly higher functional connectivity (FC, a measure of spatiotemporal synchronization) in the parietal and the parieto-occipital regions in the case of theta and alpha frequency bands and a non-significantly lower FC in the central region’s beta frequency band.”

Material and Methods

Reviewer’s Comment: Line 127. Would a broader community sample have been more useful than a possibly biased academic one?

Authors’ Response: This is in fact an important observation by the reviewer. In the current version of the manuscript, we added a new Section (Limitations and Future Direction, lines 425-495, in the current version of the manuscript) in which we discussed some of the limitations and future direction of this research. With regards to the study sample, we included the following paragraph (lines 478-486) to this Section.

“In spite of the fact that the original group of individuals who participated in our hypnosis experiment formed a moderately acceptable sample size (i.e., forty-six subjects), the final validation for their inclusion in LOW and HIGH groups based on Harvard test [63] resulted in a small sample. Furthermore, all of these individuals were university students/staff, some of whom had previous exposure to hypnosis experience. As a result, it is plausible to presume that our participants were able to more readily comprehend and follow our experimental procedure, thereby contributing to an above-average outcome that one might expect from a general population. Therefore, it is crucial to reevaluate these findings while considering a broader general population.”

Reviewer’s Comment: Line 146: Would it also have been on some research interest to also examine the EEG of the mid hypnotisability group, to see if there is a linear relationship on response differences? This may have been useful.

Authors’ Response: We discussed this matter in Section Limitations and Future Direction (lines 442-449, in the current version of the manuscript) as follows.

“Given our primary objective, we inevitably discarded a rather larger sample of individuals that were categorized as mid-hypnotizable group. However, inclusion of these individuals whose responses to hypnotic suggestions are not distinctively low or high, can potentially shed light on the nature of the observed variation of information in the brain during hypnosis. For instance, DE might be useful for determining whether the change from low to high suggestibility occurs along a continuum that encompasses the mid-suggestible group’s brain activity or such differences are rather associated with distinct and mutually exclusive neural responses.”

Reviewer’s Comment: What were the gender distribution of the remaining 14 (8L, 6H) tested? How many of these had previously been hypnotized?

Authors’ Response: Prior to providing our response to reviewer’s comment, the authors would like to clarify that the number of participants that were included in this study were 12 and not 14. The reviewer’s observation on 14 participants is correct as we originally identified 14 individuals that were not among the “Mid suggestible group” (Section Hypnosis Test and Suggestibility Score, lines 146-148, in the current version of the manuscript). However, we balanced the number of participants in such a way that LOW and HIGH groups each included 6 participants. Out of these 12 participants, four participants (one female) had previously experienced hypnosis either in form of a stage show or a research experiment. We included this information in Section Statistical Analyses (lines 217-227, in the current version of the manuscript). It reads as follows.

“Given the results of Harvard test [63], we identified a total of fourteen participants in LOW (eight participants, three females, M = 25.13, SD = 6.47) and HIGH (six participants, three females, M = 24.67, SD = 5.28) suggestible groups. First, we balanced the number of participants in HIGH and LOW suggestible groups. Result of Harvard test suggested that all the LOW suggestible participants scored either one or three. Therefore, we excluded two participants with the highest score (i.e., three in our case) at random and included the remaining six LOW suggestible participants in this group. As a result, our analyses included six participants in each of LOW (three females, M = 25.83, SD = 7.48) and HIGH groups, out of which four participants (one female) had previously experienced hypnosis either in form of a stage show or a research experiment. We adapted this selection procedure from Jiang et al. [18].”

Reviewer’s Comment: Line 151.” placed the electrodes.” How, by what scheme; electro cap should be explained, (ref?) etc?

Authors’ Response: EEG signals were recorded from 14 sites that covered the frontal, central, temporal, parietal and occipital areas. Electrodes were placed on an EEG cap (g.tec, g.GAMMAsys) according to 10-20 international system (F3, Fz, F4, T7, C3, Cz, C4, T8, P3, Pz, P4, O1, Oz, and O2) (Figure 1 (c)) and were selected to cover five main cortical regions (i.e., frontal, central, temporal, parietal and occipital) in both left and right hemispheres (red circles) and midline locations (green circles). We added this information in Section Data Acquisition, lines 171-176, in the current version of the manuscript, as follows.

“EEG signals were recorded from 14 sites that covered the frontal, central, temporal, parietal and occipital areas. Electrodes were placed on an EEG cap (g.tec, g.GAMMAsys) according to 10-20 international system (F3, Fz, F4, T7, C3, Cz, C4, T8, P3, Pz, P4, O1, Oz, and O2) (Figure 1 (c)) and were selected to cover five main cortical regions (i.e., frontal, central, temporal, parietal and occipital) in both left and right hemispheres (red circles) and midline locations (green circles).”

Reviewer’s Comment: Fig 1: were the other electrodes recorded or were just 14 recorded? If so, why were these selected? The electrode map may be misleading if the other electrodes were not used.

Authors’ Response: Only the 14 electrodes that are highlighted in Figure 1 were used in this study. We chose these electrodes due to their relative alignment with the brain regions that were identified by the previous research for their significant involvement in hypnosis: the default mode network (DMN) [64] and fronto-parietal network [1,8,10]. From a broader perspective, these electrodes covered all the major lobes of the brain that are involved in action, emotion, language, cognitive control, and action [65]. We included this information in Section Data Acquisition, lines 176-182, in the current version of the manuscript, as follows.

“We chose these electrodes due to their relative alignment with the brain regions that the previous research identified their significant involvement in hypnosis: the default mode network (DMN) [64] and fronto-parietal network [1,8,10]. From a broader perspective, the channels that were included in our study covered all the major lobes of the brain that are involved in action, emotion, language, cognitive control, and action (see [65], Chapters 9 through 12 for a detailed treatment of the subject).”

With regards to the electrodes that were used in our study, we first modified Figure 1 (c) caption to to better clarify the electrodes that we used in our study, as follows.

“Fourteen electrodes placed on the frontal, temporal, central, parietal, and occipital areas in both left and right hemispheres (red circles) and the midline locations (i.e. green circles) recorded EEG signals during the experiment.”

However, the authors would also like to state that they have no objection to change of this subplot, in case the reviewer finds it necessary.

Reviewer’s Comment: Line 176. A faster sampling rate may have been less problematic (especially for measuring higher EEG frequencies).

Authors’ Response: In the present study, we were specifically interested in behavioural responses to hypnotic suggestions that were mainly ideo-motor suggestions, inducing movement and noise in EEG. We clarified this point in Section Data Preprocessing, lines 191-195, in the current version of the manuscript. It reads as follows.

“We excluded gamma band (30-60Hz) from our analysis because the Harvard hypnosis test mainly includes motor items that require movement as a behavioural response and therefore, artefacts from muscle activity during these suggestions could have contaminated high-frequency EEG signals.”

Therefore, we decided to exclude the gamma band from our analyses, considering its vulnerability to movement related artefacts. Subsequently, we opted for a lower sampling rate of 128 Hz for EEG recordings. This choice was in accord with our overview of the EEG-based hypnosis research that identified 128 Hz and 256 Hz as the most commonly used sampling rates (references [14,31,100] in the current version of the manuscript). However, we agree with the concern of the reviewer on highlighting this limitation along with the use of high density electrodes. Therefore, we added the following discussion to Section Limitations and Future Direction, lines 467-477, in the current version of the manuscript.

“In the present study, we were specifically interested in behavioural responses to hypnotic suggestions that were mainly ideo-motor suggestions, inducing movement and noise in EEG. Therefore, we decided to exclude the gamma band from our analyses, considering its vulnerability to movement related artefacts. Subsequently, we opted for a lower sampling rate of 128 Hz for EEG recordings. This choice was in accord with our overview of the EEG-based hypnosis research that identified 128 Hz and 256 Hz as the most commonly used sampling rates (references [14,31,100] in the current version of the manuscript). However, future research that is empowered with high density electrodes and that includes higher frequency bands can allow for more comprehensive realization of the depth and breadth of brain responses to hypnotic suggestions. Such setting can also provide better testbeds for critical examination of DE and other biomarkers for study of the hypnosis.”

Reviewer’s Comment: Line 182. It would have been difficult to record gamma anyway with the low sampling rate.

Authors’ Response: Please refer to the authors’ response to reviewer’s comment “Line 176. A faster sampling rate may have been less problematic...”

Reviewer’s Comment: Line 192. Why were the rest times excluded? These could have been associated with a control baseline.

Authors’ Response: We primarily focused on determining whether DE can identify the subtle differences between high versus low suggestible individuals’ neural responses to hypnotic suggestions. Therefore, we did not include the individuals’ pre-hypnosis rest period. However, we agree with the reviewer’s point on the importance of inclusion of such baselines (e.g., as control signals) to determine whether DE can also quantify how the brain activity of these individuals differed from their respective pre-hypnosis rest time. In the current version of the manuscript, we discussed this matter in Section Limitations and Future Direction, lines 433-441, in the current version of the manuscript. It reads as follows.

“We also primarily focused on determining whether DE can identify the subtle differences between high versus low suggestible individuals’ neural responses to hypnotic suggestions. Therefore, we did not include the individuals’ pre-hypnosis rest period. Inclusion of such baselines (e.g., as control signals) can help determine whether DE can also quantify the change in the brain activity of these individuals from their respective pre-hypnosis rest time. This is indeed an important and interesting venue for future research, considering the association between hypnotizability and the brain activity during attention outside hypnosis [10] and the potential role of such a responsiveness in prediction of the individuals’ suggestibility [13].”

Reviewer’s Comment: Line 208. How was this balanced? 8 vs 6? Or another process?

Authors’ Response: Please refer to the authors’ response to reviewer’s comment “What were the gender distribution of the remaining 14 (8L, 6H) tested?”

Reviewer’s Comment: Line 222. How were 60 DE values calculated, based on time? Why 60? Needs clarifying.

Authors’ Response: Please also refer to the authors’ response to reviewer’s comment “Requires some clarity about how the 14 states ...”

Results

Reviewer’s Comment: Line 284 Should read “Change in Functional Connectivity (FC)”

Authors’ Response: The heading has been changed to “Change in Functional Connectivity (FC)”

Reviewer’s Comment: Fig 5: Clarify that the left COH map is High and right is low.

Authors’ Response: We added the following sentence to the caption of Figure 5:

“In these subplots, the left map is associated with HIGH and the right map corresponds to LOW group.”

Furthermore, we identified the subplots that corresponded to HIGH and LOW groups by adding “HIGH” and “LOW” headings to their respective subplots.

Reviewer’s Comment: It would have been interesting to see a comparison with standard coherence maps for each band with the changes in FC.

Authors’ Response: The present study was primarily meant to verify whether such information-theoretic measures as DE can benefit the hypnosis research via providing a robust quantitative measure that can distinguish between the low and high suggestible individuals. However, we also agree with the reviewer on the importance of such comparative analyses. To pinpoint the necessity for such comparative analyses, we added the following paragraph to Section Limitations and Future Direction (lines 425-432, in the current version of the manuscript).

“Our results identified DE as a potential unifying measure to reproduce previous observations that were based on multiple biomarkers. DE also appeared to further complement these measures by extending their results during the neutral hypnosis to the case of hypnotic suggestions, thereby identifying such neural activations as potential signatures of hypnotic experience. However, our results fell short in comparative analysis of DE with these previous measures (e.g., COH). Therefore, future comparative analyses of DE and these other measures to clarify their respective dis/advantages will be necessary to thoroughly appreciate their proper domain of application.”

Reviewer’s Comment: Requires some clarity about how the 14 states were used to produce the data for subsequent analyses (average of suggestion1-10?). This is mentioned in Discussion line 349. Should be addressed in Results in more detail and why?

Authors’ Response: Each individual experienced 14 phases which included a baseline recording, an induction phase (2 initial phases), 10 suggestions (10 separate phases), an awakening from hypnosis, and a post-baseline (2 final phases). The 10 suggestions in the middle where segments of interests in our study. We did not average the 10 segments, but rather computed one DE value for each segment in each frequency band and each EEG location. Given 10 suggestions and that each of HIGH and LOW groups included 6 participants, we had 6 X 10 = 60 DEs, per frequency band and for each of HIGH and LOW groups (e.g., 60 DEs for alpha band at F3). In the case of FC, we used these 60 DE values, per frequency (i.e., 6 participants × 10 suggestions), per channel, to compute the pairwise correlations among the channels. We verified this information in Section Statistical Analysis, lines 228-237, in the current version of the manuscript. It reads as follows.

“Each individual experienced 5 main stages (Figure 1) that included 14 phases: a baseline recording phase and an induction (2 initial phases), 10 suggestions (10 separate phases), an awakening from hypnosis phase, and a post-baseline (2 final phases). The 10 suggestions in the middle where segments of interests in our study. We computed one DE value for each segment in each frequency band and each EEG location. Given 10 suggestions and that each of HIGH and LOW groups included 6 participants, we had 6 × 10 = 60 DEs, per frequency band and for each of HIGH and LOW groups (e.g., 60 DEs for alpha band at F3). In the case of FC, we used these 60 DE values, per frequency (i.e., 6 participants × 10 suggestions), per channel, to compute the pairwise correlations among the channels.”

We further modified the information associated with the FC analysis (Section Change in Functional Connectivity (FC), lines 250-263, in the current version of the manuscript) to more clearly explain how DEs were used during the FC analysis. The modified Section reads as follow.

“To determine any potential significant change in functional connectivity among EEG channels of HIGH versus LOW suggestible groups, we performed all-pair FC analysis. For this purpose, we combined DEs of all participants for a given channel at a given frequency band and computed the pairwise FC using Pearson correlation (i.e., every pair of channels). This resulted in 14 × 14 FC matrices, per frequency band, where 14 refers to the number of EEG channels. For each channel, we then computed the average Pearson correlations that it had with the remainder of the channels and only considered those channels whose averaged Pearson correlations were ≥ 0.70 (i.e., primarily strong and very strong correlations) in our analysis. For the selected channels, we also counted the number of channels that they were synchronized with (i.e., number of channels that they showed ≥ 0.70 correlation with). For both of these measures (i.e., averaged correlation and number of synchronized channels, per selected channel), we applied Kruskal-Wallis test to determine the effect of suggestibility on FC. We followed this test with post-hoc paired Wilcoxon rank-sum test.”

Discussion

Reviewer’s Comment: A well written discussion about the effects/implications for the findings, however there should also be a discussion about the potential limitations of the study.

Authors’ Response: We added a new Section (Limitations and Future Direction, lines 425-495) in which we discussed some of the limitations and future direction of our research. It highlights the following topics:

1. Comparison with other measures (lines 425-432, in the current version of the manuscript): We discussed how the future can benefit from comparative analysis of DE and other measures that are used in the study of the neural correlates of hypnosis. Please refer to the authors’ response to reviewer’s comment “It would have been interesting to see a comparison with standard coherence...” for the content of the discussion that has been included in this paragraph.

2. The use of resting period EEG (lines 433-441, in the current version of the manuscript): In this paragraph, we stated the reason why we excluded the resting state EEG recordings of the participants in our study. We further pinpointed how the inclusion of this recordings in analysis of the effect of hypnosis on the brain activity can benefit the future research. for the content of the discussion that has been included in this paragraph. Please refer to the authors’ response to reviewer’s comment “Line 192. Why were the rest times excluded? These could have been associated with a control baseline.” for the content of the discussion that has been included in this paragraph.

3. The use of Mid- hypnotisable group in the future research (lines 442-449, in the current version of the manuscript): In this paragraph, we underlined the exclusion of the larger portion of our sample that corresponded to the mid-hypnotisable group, given the main objective our study. We further underlined how the inclusion of this group can help determine the potential relationship between high and low suggestible individuals. Please refer to the authors’ response to reviewer’s comment “Line 146: Would it also have been on some research interest to also examine...” for the content of the discussion that has been included in this paragraph.

4. Use of DE, minimum requirements, limitations, and potential solutions (lines 450-466, in the current version of the manuscript): In this paragraph, we summarized the our motivation for considering the entropy (in general) and highlighted the major and previous studies that brought the DE for EEG analyses to the spot light. We then briefly discussed the main assumption for the use of DE and finally highlighted how an alternative solution might be considered to lighten it. This paragraph reads as follows.

“Considering the crucial role of the cortical self-organized criticality [82–84] in maximizing its information capacity [85–87], entropy has been proven as a powerful tool for quantification of the variability in brain functioning [88] and cortical activity [89] in such broad area of research as information processing capacity of working memory (WM) [47] and the state of consciousness [91]. In this regards, although the use of DE in neuroimaging (e.g., Tononi et al. [90] and Carhart-Harris [88]) and EEG studies (e.g., Duan et al. [56], Zheng and Lu [57], and Shi et al. [58], Zheng et al. [94]), its application for modeling of the brain functioning requires further investigation. Specifically, parametric adaptation of DE for the analysis of EEG time series [56–59] assumes that the time series data under investigation is normally distributed. Although the applicability of such an assumption in neuroimaging studies has been investigated [60,94,95], similar theoretical studies to better position the use of DE in EEG-based brain research is currently (to the best of our knowledge) lacking. Such analyses can help determine the domain of applications in which DE may not be an adequate measure for modeling the EEG time series of the brain activity. Along the same direction, it is also interesting to further examine the utility of the non-parametric formulation of the differential entropy [97,98] for modeling of EEG time series of the brain signal variability [99].”

5.Sampling rate and number of electrodes in the present study (lines 467-477, in the current version of the manuscript):In this paragraph, we briefly discussed why we chose the sampling rate adapted in this study. Furthermore, we underlined the necessity for research using dense electrode EEGs and higher sampling rate for collecting the brain responses to hypnosis suggestions in higher frequencies to more comprehensively and critically examine the utility of DE and other biomarkers in hypnosis studies. Please refer to the authors’ response to reviewer’s comment “Line 176. A faster sampling rate may have been ...” for the content of the discussion that has been included in this paragraph.

6. Sample size and demographic limitations (lines 478-486, in the current version of the manuscript): This paragraph discussed the shortcomings imposed by the sample of participants that were included in our study. Please refer to the authors’ response to reviewer’s comment “Line 127. Would a broader community sample have been more useful than a possibly biased academic one?” for the content of the discussion that has been included in this paragraph.

7. Prospect of future utilization (lines 487-495, in the current version of the manuscript): We closed this Section by pointing at one potential real-world application of the findings such the results that we presented in this manuscript.

“The present findings are not only of interest to the psychology and neuroscience community but also to the researchers in the field of AI and brain-computer interfaces [101, 102]. For instance, the use of DE as a biomarker of hypnosis can be utilized in development of real-time EEG classifiers that detect their users’ responses to hypnotic suggestions. This, in turn, can expedite the deployment of the automated hypnotherapeutic systems of the future for clinical treatment of mental and behavioural disorders at brain functional level [23-26]. Such adaptations, in turn, can take the field a step closer to personalized hypnosis interventions that are tailored around the individuals’ suggestibility level.”

Reviewer’s Comment: There should also be some discussion about how the study could be expanded in future.

Authors’ Response: Please refer to the authors’ response “7. Prospect of future utilization (lines 487-495, in the current version of the manuscript)” to reviewer’s comment “Reviewer’s Comment: A well written discussion about the effects/implications for the findings…,”

Reviewer’s Comment: When is DE not an adequate biomarker? what are the bare minimum requirements to be able to calculate it? Eg Can you do a case study on an individual? Perhaps this should be discussed in Methods as well. By citing and including the work of Duan et al 2013, Wei et al 2020, Lu et al 2020(changes in emotions), Wang et al 2020 (cognitive Control), may be useful.

Authors’ Response: We discussed this matter in Section Limitations and Future Direction, lines 442-449 (Please refer to the authors’ responses to reviewer’s comment “A well written discussion about the effects/implications for the findings …,” point number 3, for the content of the discussion that has been included in this paragraph). With regards to the recommended references, we added them to the manuscript as they were in line with our study and their inclusion improved our manuscript. Specifically, Duan et al. (2013), Zheng et al. (2015), and Shi et al. (2013) were first discussed in Section Introduction, lines 96-102, in the current version of the manuscript, as follows.

“In the context of EEG time series analysis, DE appears to be first utilized by Duan et al. [56]. Subsequently, Zheng and Lu [57] noted the DE’s ability to discriminate between EEG pattern of low and high frequency, given the EEG’s higher low frequency energy over high frequency energy. They further showed (ibid.) that DE can outperform such features as differential asymmetry (DASM), rational asymmetry (RASM), and power spectral density (PSD) in EEG frequency-domain analysis. Shi et al. [58] used DE in the analysis of the EEG time series associated with vigilance.”

They were further cited in Section Limitations and Future Direction (lines 455-456, in the current version of the manuscript) during the discussion about further consideration on the use of DE.

References:

• Might be useful to add other studies cited above if used in narrative.

Authors’ Response: We cited the following references, as they benefited the clarity of our results.

56. R.-N. Duan, J.-Y. Zhu, & B.-L. Lu, Differential entropy feature for EEG-based emotion classification, Proceedings of IEEE 6TH International IEEE/EMBS Conference on Neural Engineering (NER), 7, 81-84 (2013).

57. Zheng, W. L., Lu & B. L., Investigating critical frequency bands and channels for EEG-based emotion recognition with deep neural networks, IEEE Transactions on Autonomous Mental Development, 7, 162-175 (2015).

58. Shi, L.C., Jiao, Y.Y. & Lu, B.L. Differential entropy feature for EEG-based vigilance estimation, 35th Annual International Conference of the IEEE Engineering in Medicine and Biology Society (EMBC), 6627-6630 (2013).

96. Zheng, W.L., Liu, W., Lu, Y., Lu, B.L. & Cichocki, A., Emotionmeter: A multimodal framework for recognizing human emotions, IEEE transactions on cybernetics, 49, 1110-1122 (2018).

However, we could not find Lu et al 2020(changes in emotions), Wang et al 2020 (cognitive Control). We would like to ask the reviewer to provide the DOIs of these articles, in case the reviewer finds their inclusion may further benefit our manuscript.

The other newly cited research in the current version of the manuscript are references 82-97 and 100-102.

Responses to Reviewer 2

Reviewer #2: Page 1 (Abstract)

Reviewer’s Comment: Change “Variational information” to “Variation of information”

Authors’ Response: All occurrences of “Variational information” are changed to “Variation of information”

Reviewer’s Comment: Change “accompanies with a” to “accompanied by a”.

Authors’ Response: All occurrences of “accompanies with a” are changed to “accompanied by a”

Reviewer’s Comment: Change “provides a direct” to “provides direct”.

Authors’ Response: “provides a direct” is changed to “provides direct”

Reviewer’s Comment: Page 2 lines 34 and 35

The phrase “power and information do not necessarily need to be modulated” is confusing in the context of the wider sentence. I suspect the authors intend to convey either that modulation of power does not necessarily involve modulation of information or that modulation of information does not necessarily involve change in local power or BOTH. The sentence should be rewritten to better convey the authors intended meaning.

Authors’ Response: We in fact intended the second description that is mentioned by the reviewer. Therefore, we modified the manuscript by replacing our sentence with “modulation of information does not necessarily involve change in local power.” In the current version of the manuscript, it reads as follows (Section Introduction, lines 34-36, in the current version of the manuscript).

“Wutz et al. [39] pointed that the modulation of information does not necessarily involve change in local power, thereby implying the possibility of the presence of a significant information when power is not elevated.”

Reviewer’s Comment: Page 3 line 68 “significant differences” should probably be changed to read “significant differences in power”.

Authors’ Response: “significant differences” is changed to “significant differences in power”

Reviewer’s Comment: Line 73 change “perspective” to “perspectives” and change “as role” to “as the role”.

Authors’ Response: “perspective” is changed to “perspectives” and “as role” is changed to “as the role”

Reviewer’s Comment: Page 6 line 180 “major artefacts” is simply too vague and in-descriptive consider changing this to something like “gross movement artefacts”.

Authors’ Response: “major artefacts” is changed to “gross movement artefacts”

Results

Reviewer’s Comment: Page 8 figure 3 legend P<.05 and p<.01 are un-necessary as there are no such results displayed.

Authors’ Response: P<.05 and p<.01 are removed from Figure 3.

Reviewer’s Comment: Numerical values presented in Table 1 should not be reproduced in text on page 8 and page 10.

Authors’ Response: Numerical values in text on pages 8 and 10 are removed.

Reviewer’s Comment: The values shown in Table 1 appear to be inconsistent with the corresponding images displayed in Figure 3 for the means for lows and highs for beta at P3 and also for alpha at O2. Please check this. Consider if Figure 3 adds value for the reader in understanding these results.

Authors’ Response: We checked all the values in Table1 and verified that they were correct. With regards to Figure 3, the authors believe that its inclusion allows the readers to visually appreciate the significant differences that have been reported in this manuscript. However, the authors are willing to exclude the figure in case the reviewer considers this modification to be necessary.

Reviewer’s Comment: Page 11 line 287 presents an eta squared value as does line 296. Please check is eta squared the intended/appropriate/correct effect size measure?

Authors’ Response: We checked these values. Whereas the eta squared at line 287 (line 304, in the current version of the manuscript) is 0.28, it is 0.08 in line 296 (line 312 in the current version of the manuscript). The authors would also like to bring to reviewer’s kind attention that whereas the first eta-squared is associated with Kruskal-Wallis test on FC values, the latter pertains to this applied on the number of channels that different channels were synchronized with (i.e., the number of undirected connections among them).

Reviewer’s Comment: Repetition of numerical values from table 2 and table 3 in text on page 11 (and page 12) appears redundant and un-necessary. This is probably best removed.

Authors’ Response: Numerical values from table 2 and table 3 in text are removed.

Discussion

Reviewer’s Comment: Page 12 Line 329 refers to findings in separate hemispheres. This seems to imply hemisphere specific tests were conducted and reported but I do not recall seeing such. Please clarify this issue for the readers.

Authors’ Response: We rewrote this sentence as follows (Section Discussion, lines 340-343, in the current version of the manuscript).

“We observed that the large effect of hypnotic suggestibility on information content of the theta, alpha, and beta frequency bands was not confined to the EEG channels that covered a specific hemisphere but manifested (with comparable strength in their effect sizes) in both, EEG channels on the left as well as the right hemispheres.”

Reviewer’s Comment: Page 13. Lines 348 – 349 “did not identify any significant differences” did you mean “differences in alpha”? If so please state that or otherwise clarify the frequency bands referred to.

Authors’ Response: We rewrote this part (Section Discussion, lines 358-363, in the current version of the manuscript) as follows.

“In this respect, Jamieson and Burgess [40] also reported similar changes in FC from pre-hypnosis to hypnosis state using iCOH (increase in theta) [44] and COH (decrease in beta1) [43]. However, their analyses which were primarily based on the state of hypnosis (i.e., without observing responses of the participants to hypnotic suggestions) did not identify any significant differences [40] between pre-hypnosis and hypnosis state on these bands.”

Reviewer’s Comment: Line 370 change “the hypnosis” to “hypnosis”.

Authors’ Response: All occurrences of “the hypnosis” are changed to “hypnosis”

Reviewer’s Comment: Page 14 line 392 change “to speculate” to read “to propose that”.

Authors’ Response: “to speculate” is changed to “to propose that”

Reviewer’s Comment: Line 402 reference [77] consider adding an additional recent relevant finding on the role of functional connectivity in upper alpha in hypnotic amnesia suggestion responses

Jamieson, G. A., Kittenis, M. D., Tivadar, R. I., & Evans, I. D. (2017). Inhibition of retrieval in hypnotic amnesia: dissociation by upper-alpha gating. Neuroscience of consciousness, 3(1).

Note my conflict of interest I am a co-author. Please do not include this reference here unless you consider it adds value for the reader.

Authors’ Response: Thank you very much for your suggestion and the clarity on issues potentially associated “conflict of interest.” We cited (Section Discussion, Reference No. 82, line 416, in the current version of the manuscript) this reference as we found its findings to be useful to the potential readers interested in the line research presented by our results and other results cited in our manuscript.

Responses to Editor

Editor’s Comment: 2. Please provide additional details regarding participant consent. In the ethics statement in the Methods and online submission information, please ensure that you have specified whether consent was informed.

Authors’ Response: The Ethics Statement is moved to Section Materials and Methods, lines 269-272, in the current version of the manuscript. The current statement also provides additional information regarding the “informed written consent” from the participants. It reads as follows.

“All subjects singed a written informed consent from in accordance with ethical approval of the Ethics Committee (Approval number: 412-2), University of Tokyo. Every participant received a payment at the end of the experiment.”

Editor’s Comment: b) If there are no restrictions, please upload the minimal anonymized data set necessary to replicate your study findings as either Supporting Information files or to a stable, public repository and provide us with the relevant URLs, DOIs, or accession numbers. For a list of acceptable repositories, please see http://journals.plos.org/plosone/s/data-availability#loc-recommended-repositories.

Authors’ Response: Upon acceptance of the manuscript, data-files of the extracted entropy-features that have been utilized during analyses reported in this article will be uploaded in a public repository by the second and third authors of this manuscript.

---

## [Editor Report · Decision Letter 1]

11 Mar 2020

Higher Hypnotic Suggestibility Is Associated with the Lower EEG Signal Variability in Theta, Alpha, and Beta Frequency Bands

PONE-D-20-00814R1

Dear Dr. Keshmiri,

We are pleased to inform you that your manuscript has been judged scientifically suitable for publication and will be formally accepted for publication once it complies with all outstanding technical requirements.

With kind regards,

Vilfredo De Pascalis

Academic Editor

PLOS ONE
---

## [Editor Report · Acceptance letter]

17 Mar 2020

PONE-D-20-00814R1

Higher Hypnotic Suggestibility Is Associated with the Lower EEG Signal Variability in Theta, Alpha, and Beta Frequency Bands

Dear Dr. Keshmiri:

I am pleased to inform you that your manuscript has been deemed suitable for publication in PLOS ONE. Congratulations! Your manuscript is now with our production department.

With kind regards,

on behalf of

Prof. Vilfredo De Pascalis

Academic Editor

PLOS ONE